# Availability of essential, generic medicines before and during COVID-19 at selected public pharmaceutical supply agencies in Ethiopia: a comparative cross-sectional study

Tsegaye Melaku [1], Zeleke Mekonnen,[1] Gudina Terefe Tucho [1], Mohammed Mecha,[1] Christine Årdal,[2] Marianne Jahre[3]

TM and ZM contributed equally.

[1]Institute of Health, Jimma University, Jimma, Ethiopia
[2]Norwegian Institute of Public Health, Oslo, Norway
[3]BI Norwegian Business School, Oslo, Norway

**Correspondence to**
Tsegaye Melaku;
tsegish.melaku@gmail.com

## ABSTRACT

**Objectives** Lockdowns and border closures impacted medicine availability during the COVID-19 pandemic. This study aimed to assess the availability of essential, generic medicines for chronic diseases at public pharmaceutical supply agencies in Ethiopia.

**Design** Comparative cross-sectional study.

**Setting** The availability of essential, generic medicines for chronic diseases was assessed at two public pharmaceutical supply agency hubs.

**Participants** The current study included public supply agency hub managers, warehouse managers and forecasting officers at the study setting.

**Outcomes** The assessment encompassed the availability of chronic medicines on the day of data collection, as well as records spanning 8 months before the outbreak and 1 year during the pandemic. A total of 22 medicines were selected based on their inclusion in the national essential drug list for public health facilities, including 17 medicines for cardiovascular disease and 5 for diabetes mellitus.

**Results** The results of the study indicate that the mean availability of the selected basket medicines was 43.3% (95% CI: 37.1 to 49.5) during COVID-19, which was significantly lower than the availability of 67.4% (95% CI: 62.2 to 72.6) before the outbreak (p<0.001). Prior to COVID-19, the overall average line-item fill rate for the selected products was 78%, but it dropped to 49% during the pandemic. Furthermore, the mean number of days out of stock per month was 11.7 (95% CI: 9.9 to 13.5) before the outbreak of COVID-19, which significantly increased to 15.7 (95% CI: 13.2 to 18.2) during the pandemic, indicating a statistically significant difference (p<0.001). Although the prices for some drugs remained relatively stable, there were significant price hikes for some products. For example, the unit price of insulin increased by more than 130%.

**Conclusion** The COVID-19 pandemic worsened the availability of essential chronic medicines, including higher rates of stockouts and unit price hikes for some products in the study setting. The study's findings imply that the COVID-19 pandemic has aggravated already-existing medicine availability issues. Efforts should be made to develop contingency plans and establish mechanisms

## STRENGTHS AND LIMITATIONS OF THIS STUDY

⇒ Provides a comparative assessment of selected generic medicine's availability used for chronic diseases before and during COVID-19.
⇒ The quality of the data in our study is high as it uses a standardised methodological tool recommended by WHO.
⇒ Limited geographic location will affect the study representativeness.

to monitor medicine availability and pricing during such crises.

## INTRODUCTION

Medicines are vital elements to healthcare, and access to medicines is a fundamental human right.[1] WHO defines essential medicines as those which 'satisfy the population's priority healthcare needs'.[2] Drug shortages are commonplace and present significant problems in all countries, including developed nations. The causes of drug shortages are complex and multifaceted, and addressing them requires a collaborative effort from various stakeholders. The majority of the developing countries, mostly in sub-Saharan Africa, lack consistent availability and access to essential health products.[3 4] This is due to various factors including inadequate infrastructure, insufficient funding, currency fluctuations and more system-related factors like poor governance and corruption.[5–10]

Although various emergency situations and supply contamination have shut down key manufacturing sites in the past, the COVID-19 pandemic was the first sustained shock event to affect all parts of the global pharmaceutical supply chain.[11 12] During the pandemic, countries across the world went into lockdowns,

BMJ

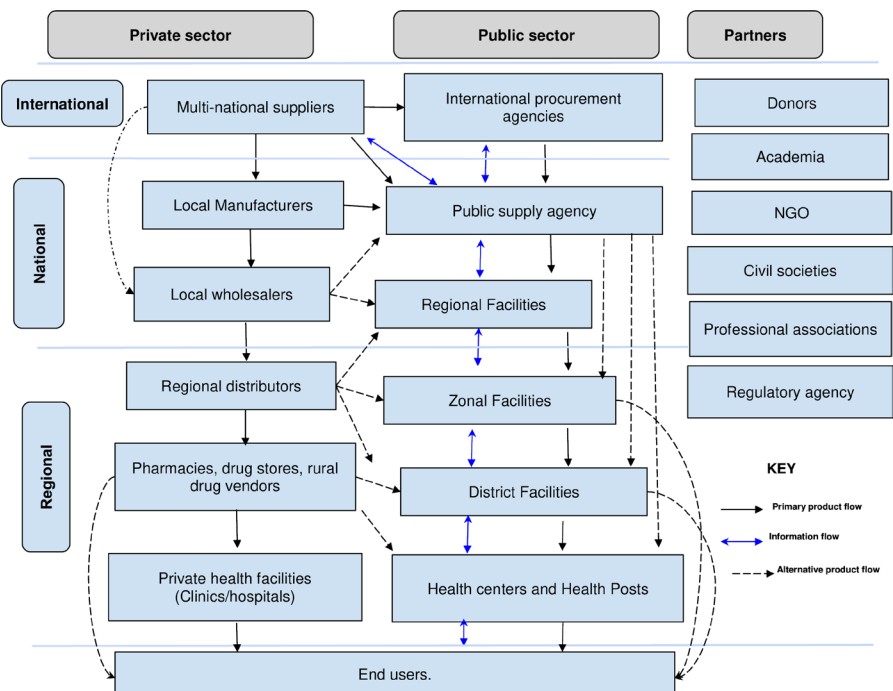

**Figure 1** Pharmaceutical supply chain map of ethiopian pharmaceutical market (adapted from management sciences for health 2012.[38]

shutting down or reducing transport within and between them. This affected the manufacturing, supply and distribution of medicines, leading to constraints in the medicine supply chain. A general shortage of various types of pharmaceuticals at supply and retail levels has been recorded in many nations, along with price rises and increases in falsified products.[13 14] The impact of the COVID-19 pandemic on the global medical supply chain has been exacerbated by core weaknesses in relation to essential medicine production and procurement.[15 16]

Drug shortages caused by the COVID-19 outbreak were mostly fuelled by spikes in demand for certain kinds of medications such as medicines used for the management of non-communicable diseases (NCDs).[16–18] People with NCDs need uninterrupted, reliable access to quality-assured and affordable medicines and health products. During the pandemic, service delivery points experienced shortages of medications used to treat cardiovascular disease (CVD) and diabetes mellitus (DM).[18 19] Most low-income governments, especially those in sub-Saharan Africa, opt for a distribution strategy where they purchase medicines and deliver them to health facilities via a publicly run Central Medical Store and a government-owned transport fleet. Recent research, which assessed the availability of the same basket of NCD medicines at the health facility level, showed that COVID-19 significantly increased shortages of these products and worsened the stockout situation at the health facilities.[20] Expanding on its antecedent, the present manuscript delineates and scrutinises the impact of the COVID-19 pandemic on the supply chain dynamics of NCD medications at the level of pharmaceutical supply agencies, immediately preceding their distribution to healthcare facilities.

## METHODS
### Study setting
The study was conducted at two Ethiopian pharmaceutical supply agency (EPSA) hubs (Nekemte and Jimma) in southwestern Ethiopia. Both hubs, Nekemte and Jimma, are approximately 300 kilometres and 350 kilometres from Addis Ababa, respectively, and about 200 kilometres from each other. According to the current supply agency clustering scheme, these two hubs are under western cluster, which is one of the seven clusters consisting of three hubs (Jimma, Nekemte and Gambella). The agency hubs serve as outposts to distribute pharmaceuticals, chemical reagents, medical supplies and equipment procured by the central office to localities on all corners of the country/region. They also gather pharmaceutical demand within their domain and communicate it with the head office to form an aggregate demand at a national level. Hubs have been serving as key governmental structures for the implementation of integrated pharmaceutical and logistic services (IPLSs) and the distribution of essential health commodities for public health facilities. They also provide supervisory, material support and capacity building to health facilities for strengthening and enforcing the implementation of IPLS and pharmacy services.[21] The pharmaceutical supply chain in Ethiopia consists of public and private actors and also partners where some of the manufacturing is done in country. This system mainly comprises manufacturers/suppliers, wholesalers and/or distributors to the health facilities as the main actors in the pharmaceutical supply chain map (figure 1).

### Sample selection and time periods

22 essential, generic medicines were selected from the essential medicine list of the country. 17 were CVD drugs, and the remaining 5 were antidiabetics. The products with the lowest dosage strength and available formulation of the substance, which have a market authorisation, were included. These are the most commonly prescribed medicines and dosages in the health facilities in Ethiopia. The LMIS records were used to track and compare product transaction data for 8 months before COVID-19 (between 1 May and 31 December 2019) and for 12 months during the pandemic time (between 1 January and 31 December 2020). This period was purposefully selected. In Ethiopia, the month of May/June marks the culmination of fiscal year activities by the government and the preparation of the New Year budget. Consequently, we have only 8 months counting back from 31 December 2019 (the outbreak of COVID-19) to obtain the most up-to-date information about the stock.

### Data collection procedure

Based on the study objective, the modified data collection checklist from the standard tools[22 23] was used to collect the relevant data. Closed ended questions about the quantity of medicines ordered, received and distributed and the stock situation on the day of data collection were included on the observation checklist. Data were collected through a cross-sectional survey conducted over a 12-month period of time during the COVID-19 pandemic. The researchers visited pharmaceutical supply agencies and collected information on the availability of NCD medicines. They also recorded any shortages or stockouts that occurred during this period. A 2-day training session was given to the data collectors regarding the study's objectives and how to collect the data. During the data collection process, there was close supervision by the investigators to ensure consistency and completeness of the data. The data were collected from the two public supply agency hubs.

### Data processing and analysis

A descriptive frequency and percentage analysis was conducted to evaluate the line-item fill rate, product availability and distribution trends before and after the COVID-19 pandemic. The following formulas were used to estimate the current product availability, stockout duration and line-item fill rate of the selected medicines.

$$Medicine\ availability = \frac{Number\ of\ product\ available\ in\ a\ given\ period}{Number\ of\ products\ considered} \times 100$$

$$Line\text{-}item\ fill\ rate = \frac{Amount\ (quantity)\ of\ item\ received}{Amount\ (quantity)\ of\ item\ ordered} \times 100$$

$$Average\ stock\text{-}out\ duration = \frac{Number\ of\ days\ the\ product\ was\ not\ available}{Number\ of\ products\ reviewed} \times 100$$

Based on the WHO's drug availability index, the analysis considered the availability of products that are ordered and received in the supply agencies as well as stock status on the day of data collection. A product is available if it is in stock in the health facility providing the service on the day of the visit or during the specified period. The following ranges were used to describe the medication availability status: <30% is very low, 30%–49% is low, 50%–80% is fairly high and >80% is very high.[24 25]

### Patient and public involvement

It was not appropriate or possible to involve patients or the public in the design, conduct, reporting or dissemination plans of our research.

## RESULTS
### The line-item fill rate

The line-item fill rate indicates how often the EPSA hubs received at least 80% of the quantity ordered for a particular medicine. Prior to COVID-19, the overall average

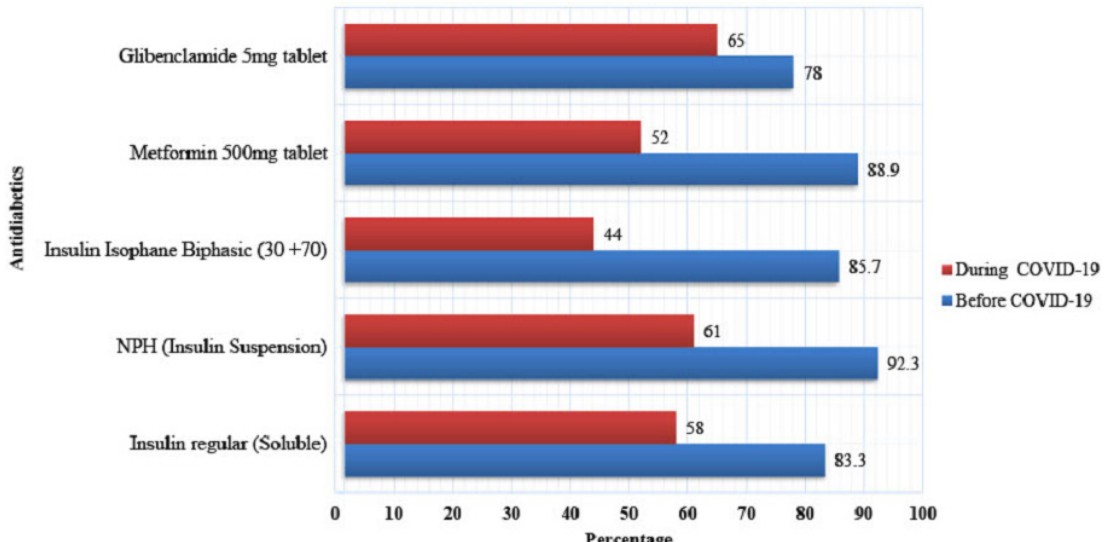

**Figure 2** The average line-item fill rate for selected antidiabetics before and during COVID-19.

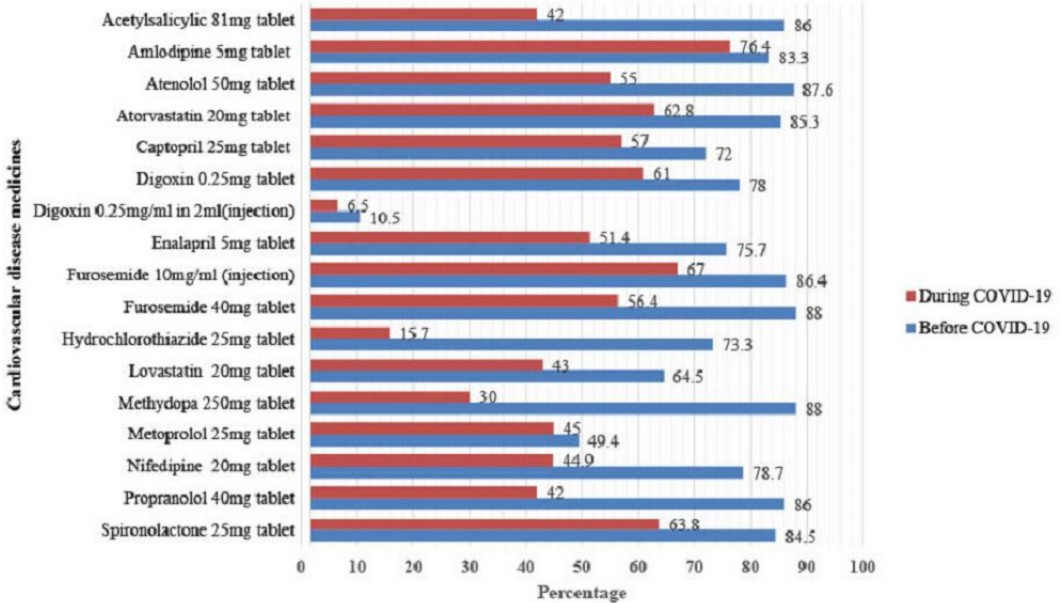

**Figure 3** The average line-item fill rate for selected CVD medications before and during COVID-19.

**Table 1** Product availability and average days out of stock for selected antidiabetics and CVD medications before and during COVID-19 pandemic

| List of drug products | Prepandemic | | During the pandemic | | Decline in availability (%) |
|---|---|---|---|---|---|
| | Average days out of stock per month | Availability* (%) | Average days out of stock per month | Availability* (%) | |
| Insulin regular (soluble) | 9.5 | 74.7 | 13.3 | 44.6 | 40 |
| NPH insulin (suspension) | 8 | 77.3 | 10.9 | 49 | 37 |
| Insulin isophane (biphasic) | 8.5 | 75.2 | 12.1 | 43.4 | 42 |
| Metformin 500 mg tablet | 8.7 | 76.5 | 12.0 | 47.3 | 38 |
| Glibenclamide 5 mg tablet | 10.3 | 70.6 | 12.3 | 56.9 | 19 |
| Acetylsalicylic acid 81 mg tablet | 9.2 | 70.8 | 12.7 | 44 | 38 |
| Enalapril 5 mg tablet | 11.0 | 68 | 14.4 | 47 | 31 |
| Captopril 25 mg tablet | 16.7 | 56 | 26 | 23 | 59 |
| Atenolol 50 mg tablet | 9.6 | 73.7 | 11.8 | 56.4 | 24 |
| Propranolol 40 mg tablet | 9.4 | 72.6 | 13.2 | 44 | 39 |
| Metoprolol 25 mg tablet | 19 | 52 | 21.2 | 46 | 12 |
| Furosemide 40 mg tablet | 9.3 | 71.5 | 12.4 | 47.7 | 33 |
| Furosemide 10 mg/mL | 8.4 | 76 | 10 | 61.5 | 19 |
| Hydrochlorothiazide 25 mg tablet | 10.8 | 68 | 16 | 35 | 49 |
| Spironolactone 25 mg tablet | 9.6 | 74 | 11 | 63.5 | 14 |
| Atorvastatin 20 mg tablet | 15 | 62 | 20 | 41.5 | 33 |
| Lovastatin 20 mg tablet | 17 | 45 | 28 | 10.5 | 77 |
| Methyldopa 250 mg tablet | 11.3 | 66.8 | 17.2 | 32 | 52 |
| Amlodipine 5 mg tablet | 8.7 | 78 | 11.2 | 56 | 28 |
| Nifedipine 20 mg tablet | 13.5 | 68.3 | 19.5 | 38 | 44 |
| Digoxin 0.25 mg tablet | 9.6 | 74 | 12.4 | 52.5 | 29 |
| Digoxin 0.25 mg/mL (injection) | 24 | 31 | 27 | 13 | 58 |

*Based on the WHO availability index.[24 25]
NPH, Neutral Protamine Hagedorn.

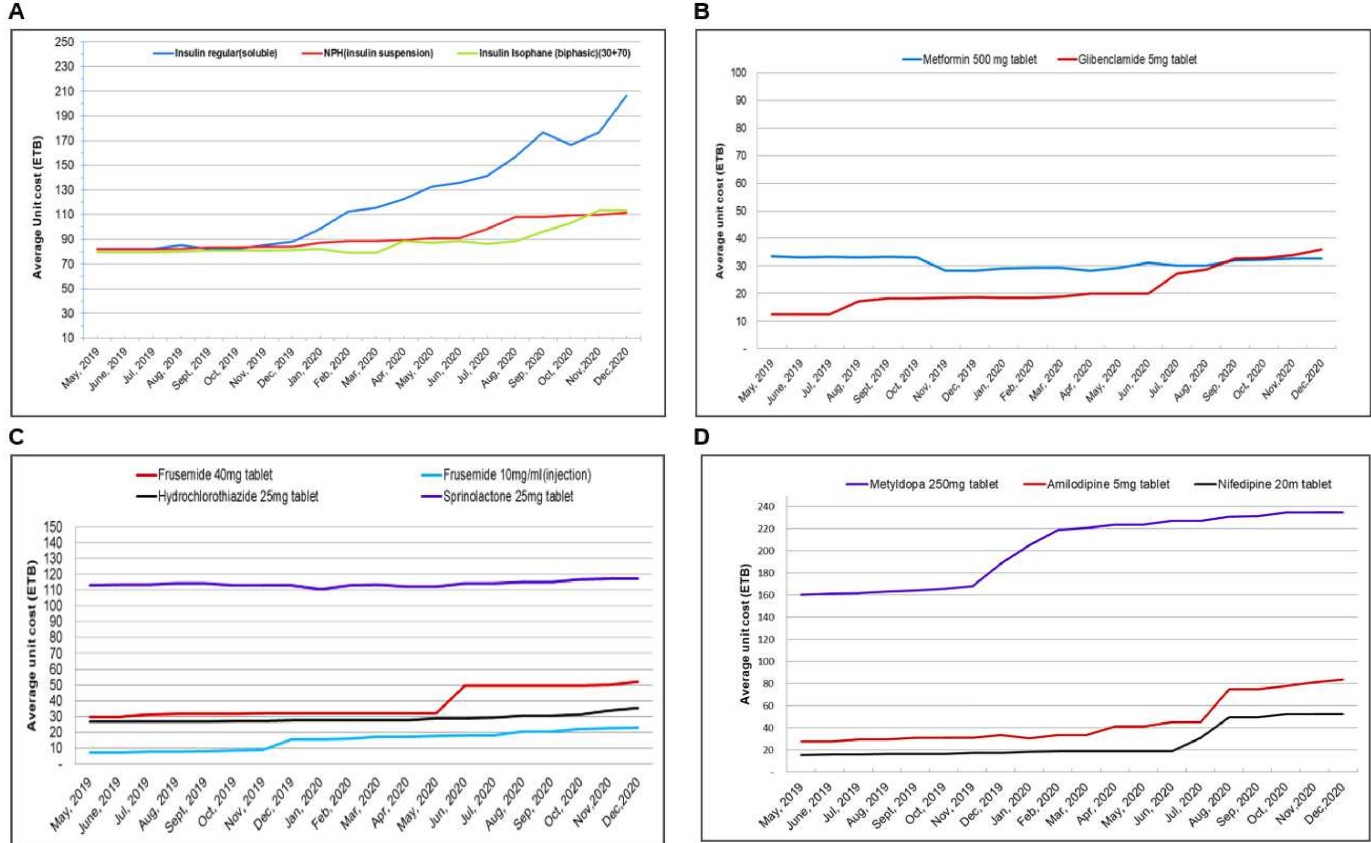

**Figure 4** Trends of selected antidiabetics (A, B) and CVD medicine (C, D) unit prices at public pharmaceutical supply agency hub.

line-item fill rate for the selected products was 78%. It declined to 49% during the pandemic. For antidiabetics, the average line-item fill rate declined from 85% to 56%. For CVD drugs, it went down from 76% to 46%. The overall line-fill rate dropped for all selected medicines during COVID-19 (figures 2 and 3).

### Product availability and stockout situation of selected basket medicines

During COVID-19, the overall mean (95% CI) of availability of the selected basket medicines was 43.3% (37.1 to 49.5). However, it was 67.4% (62.2 to 72.6) before the outbreak. Comparing the mean availability of the two periods using a two-sample t-test, it showed a statistically significant difference (p<0.001). On the day of data collection, 10 of the 22 (46%) were out of stock.

Concerning the days out of stock, the mean (95% CI) was 11.7 (9.9 to 13.5) days before COVID-19 outbreak, which was increased to 15.7 (13.2 to 18.2) days. This showed a statistically significant difference (p<0.001). Before the pandemic, the average stockout duration per month was 9 days for antidiabetics and 11 days for the CVD drugs. This increased to 13 days and 17 days/month, during the pandemic for antidiabetics and the CVD drugs, respectively. The medicines with the least supply were captopril 25 mg tablets, metoprolol 25 mg tablets, lovastatin 20 mg tablets and digoxin 0.25 mg/mL injection. Lovastatin had the longest average stockout of 28 days, closely followed

by digoxin injection at 27 days and captopril tablet at 26 days (resulting in WHO availability indices of 11%, 13 % and 23%, respectively). The drop in the availability of specific medications ranged from 12% to 77% during the pandemic (table 1).

### Unit price changes of selected antidiabetics and CVD medications

We were unable to assess the unit pricing changes of all medicines because of stockouts for the majority of the products. The prices were relatively stable for some medicines (where there were sufficient data to assess). Most of these products were from the local pharmaceutical industries. Yet for some of the products, there were dramatic price increases (figure 4), such as for insulin products and methyldopa. Some of these products were imported and supplied by international pharmaceutical suppliers such as Novo Nordisk Pharmatech A/S. Likewise, the unit price for oral furosemide, amlodipine and nifedipine increased starting from June 2020. During the study period, the value of the Ethiopian birr decreased by 38% compared with US dollar (USD),[26] dramatically increasing the price of imported medicines.

### Selected medicine manufacturer/supplier

This paper uses the terms manufacturers and suppliers interchangeably. Sometimes facilities supply their stock from importer and/or wholesaler and sometimes directly

**Table 2** Lists of medicine manufacturer/supplier during and prior to the pandemic

| S. no | Manufacturers/supplier name (country) | Percentage of supply orders covered | |
| --- | --- | --- | --- |
| | | Before COVID-19 | During COVID-19 |
| 1. | Amino AG (Switzerland) | 4 | 8 |
| 2. | Addis Pharmaceutical Factory (Ethiopia) | 8 | 13 |
| 3. | AstraZeneca (UK) | 4 | 4 |
| 4. | Bliss GVS Pharma Limited (India) | – | 3 |
| 5. | Brawn Laboratories Limited (India) | 4 | 1 |
| 6. | Cadila Pharmaceuticals PLC (Ethiopia) | 10 | 10 |
| 7. | Ciron Drugs & Pharmaceuticals PLC (India) | 4 | 4 |
| 8. | East African Pharmaceuticals PLC (Ethiopia) | 6 | 10 |
| 9. | EIPICO (Egypt) | 4 | 4 |
| 10. | Epharm (Ethiopia) | 5 | 1 |
| 11. | Gulf Pharmaceuticals Ltd (UAE) | 4 | 4 |
| 12. | Humanwell Pharmaceutical PLC (Ethiopia) | 4 | 1 |
| 13. | Intas Pharmaceuticals (India) | 2 | 1 |
| 14. | Julphar Pharmaceuticals PLC (Ethiopia) | 8 | 5 |
| 15. | Laboratoires Sterop (Belgium) | 2 | 1 |
| 16. | Novo Nordisk Pharmatech A/S (Denmark) | 4 | 5 |
| 17. | Remedica Ltd (Cyprus) | 13 | 13 |
| 18. | Reyoung Pharmaceutical Co., Ltd (China) | 4 | 1 |
| 19. | Sansheng Pharmaceutical PLC (Ethiopia) | 9 | 13 |

EIPICO, Egyptian International Pharmaceutical Industries Company; Epharm, Ethiopian Pharmaceuticals Manufacturing share company.

from the manufacturer. For the selected NCD medicines, there were 19 manufacturers/suppliers over the period. During the 8 months prepandemic, 192 orders were placed by the 2 agencies, while during the 12 months of the pandemic, 264 supply orders were placed. Ethiopian companies represented 50% and 53% of suppliers before and during the COVID-19 pandemic, respectively. Similarly, European pharmaceutical industries were supplying 27% prior to COVID-19 and 31% during the pandemic, meaning that about 84% of NCD medicines at the two agencies were supplied either from Ethiopia or Europe during COVID-19 (table 2).

## DISCUSSION

Ensuring sustainable availability of essential medicines has been a persistent challenge in Ethiopia. The country has made progress in recent years, but the COVID-19 pandemic highlighted the need for further improvements. More must be done to strengthen the drug supply chain and ensure sustainable availability of essential medicines. This study assessed trends in the availability of selected medicines for chronic diseases at the public supply agencies over a long time period. We evaluated the fill-item rate, trends and product availability before and during the pandemic.

The results of the study showed that the COVID-19 pandemic significantly increased shortages of NCD medicines at pharmaceutical supply agencies. The researchers found that the stockout situation worsened during the pandemic, indicating challenges in the supply chain of essential medicines. Some of the challenges include disruptions in transportation and logistics, shortages of raw materials and active pharmaceutical ingredients, increased demand for certain medications and workforce shortages due to illness or quarantine measures. Additionally, border closures and export restrictions have also impacted the availability of essential medicines in some regions. In response to the global burden of NCDs, WHO has developed a Global Action Plan that includes a target of 80% availability and affordability of essential medicines for the prevention and treatment of DM, CVD and respiratory diseases.[27] Despite this strong recommendation, the availability of NCD medicines in Ethiopia falls below the recommended target. At the national level, the median availability of medicines for NCDs was below 50% with average stockout days of up to 27 days.[28] Similarly, the current study showed that the overall availability of selected medicines was 49% during the pandemic, down from 78% prepandemic. This includes wide variation, with lovastatin available only 11% according to the WHO availability index during the pandemic. This is consistent with studies from Kenya,[29] Portugal,[30] Poland[31] and Congo and Cameroon.[32] This has led to increased difficulties for individuals with chronic conditions to access the medications they need to manage their health. The higher rates of stockouts mean that pharmacies and

healthcare facilities are unable to meet the demand for these essential medicines, leaving patients without access to necessary treatments.

Not only were the medicines less available during COVID-19, but some also became more expensive, which is in line with several studies from countries like Bangladesh,[33] Portugal,[30] Saudi Arabia,[34] India,[35] Nigeria[36] and Namibia.[37] While the Ethiopian birr decreased in value compared with the USD by 38%, soluble insulin increased by more than 130% in price, which means exchange rates cannot alone explain the price increase. This suggests that other factors, such as supply chain disruptions or increased production costs, may have contributed to the price hike. Moreover, the changes in supplier dynamics on medication availability and affordability might contribute to the price changes during the pandemic. These findings highlight the need for further investigation into the underlying causes of medication price increases during crises. It is crucial to understand the specific factors influencing these price changes to develop effective strategies for mitigating their impact on individuals with chronic conditions. This study also suggests that while there were stable prices for certain medicines supplied by local pharmaceutical industries, there were price increases for insulin products and methyldopa, which were imported and supplied by international pharmaceutical suppliers. Factors such as supply chain disruptions, increased demand and changes in production costs could contribute to these price fluctuations. It is important to further investigate the reasons behind these price increases, as they may have implications for medication affordability and accessibility.

Furthermore, this study also assessed the type of suppliers or manufacturers of selected drugs before and during the epidemic. The supply distribution did not vary substantially by region during and after COVID-19. For NCD medicines, it appears that at least these two public pharmaceutical supply agencies are mainly supplied by Ethiopian and European companies. The large proportion of Ethiopian suppliers is beneficial, likely both in normal times and in pandemic, as the currency fluctuations should be smaller, although still present due to a likely dependence on imported active pharmaceutical ingredients (APIs). That local pharmaceutical industries were able to step up and meet the demand for these products demonstrates the importance of having strong and resilient local pharmaceutical industries that can provide essential healthcare products during times of crisis. This study highlights the importance of addressing supply chain disruptions during public health emergencies, such as the COVID-19 pandemic, to ensure the availability of essential medicines for patients with NCDs. It emphasises the need for interventions and strategies to strengthen the pharmaceutical supply chain and improve access to essential medicines in low-income countries. Overall, the COVID-19 pandemic has underscored the critical importance of maintaining a robust and resilient healthcare system that can effectively respond to emergencies while ensuring the availability and affordability of essential medicines for those who need them most.

## Limitation of the study
To the best of the authors' knowledge, this is one of the first study assessing supply situations of essential medicines during a pandemic at selected public pharmaceutical supply agencies. Yet, the results may not apply to all medicines. Other therapeutic areas may experience unique trends. The study included only two facilities, which may have unique characteristics.

## CONCLUSION
In summary, significant shortages of selected basket medicines used for the management of DM and CVD were observed at the two selected public pharmaceutical supply agencies. The pandemic has highlighted the importance of ensuring access to essential medications for patients with chronic conditions such as CVD and diabetes. Unit prices of some of the medicines dramatically increased, due to both currency fluctuations and other possible factors. The supply distribution did not vary substantially by region during and after COVID-19, with the two agencies mainly being supplied by Ethiopian and European companies. The findings of this study suggest that the COVID-19 pandemic has exacerbated existing medicine availability problems. Yet it also demonstrates a strong local provision of NCD medicines. Challenges facing these local providers should be further studied to understand how currency fluctuations, dependencies on imported APIs and other factors influence their ability to supply the local population. It would also be useful to compare these findings with patient experiences to better understand how these stockouts impacted them and their health.

**Contributors** ZM, TM, GTT, MM, CÅ and MJ designed the study. ZM and TM drafted the first version of the manuscript. ZM, TM, CÅ and MJ discussed the analytic strategy. ZM, TM and GTT did the data curation and performed the analysis. All authors reviewed the final paper and made creative and substantial contributions, approved the final version and consented to publication. TM is the overall guarantor, and access to all the analysis codes and datasets.

**Funding** This work was supported by the Global Health and Vaccination Research Programme, which is the programme of the Research Council of Norway.

**Competing interests** None declared.

**Patient and public involvement** Patients and/or the public were not involved in the design, or conduct, or reporting, or dissemination plans of this research.

**Patient consent for publication** Not applicable.

**Ethics approval** This study involves human participants and was approved by the Institutional Review Board of the Institute of Health, Jimma University in Ethiopia (reference number JHRPG/1043/2020). Participants gave informed consent to participate in the study before taking part.

**Provenance and peer review** Not commissioned; externally peer reviewed.

**Data availability statement** Data are available upon reasonable request. This study included all pertinent data. Additional information could be obtained from the corresponding author upon reasonable request.

**ORCID iDs**
Tsegaye Melaku http://orcid.org/0000-0001-8373-5309
Gudina Terefe Tucho http://orcid.org/0000-0001-7848-5456

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
