## [Reviewer comments · BMJ Open]

ARTICLE DETAILS

TITLE (PROVISIONAL)	Availability of essential, generic medicines before and during COVID-19 at selected public pharmaceutical supply agencies in Ethiopia: Comparative cross-sectional study
AUTHORS	Melaku , Tsegaye; Mekonnen, Zeleke; Terefe Tucho, Gudina; Mecha, Mohammed; Årdal, Christine; Jahre, Marianne

VERSION 1 – REVIEW

REVIEWER	Khuluza, Felix University of Malawi, College of Medicine-Pharmacy Department
REVIEW RETURNED	29-Aug-2023

GENERAL COMMENTS	Thank you for a great article that has compared drug availability before and during COVID-19 pandemic. They are few articles in this field and will enhance measures aimed at promoting drug availability. However the authors need to address the following: 1. Paragraph two of the introduction section: Please replace the word "counterfeit". This word is no longer recommended by WHO as it is property rights issue. And use the word "falsified" as advocated by WHO.2. diagram 1 is a general distribution system of Ethiopia. I would rather request the authors highlights the hubs under which the study took place than a picture of the whole country.3. Diagram 2. the diagram is slightly confusing in the following ways:-It is assuming that Donors do manufacture medicines. this is not entirely true as in most cases they just finance the purchase of medicines from reputable supplies. So the diagram has combined flow of product and funding. It needs to be corrected.-What are the meaning of the arrows in diagram 2? I see nine arrows from API to manufacturers and four from Manufacturers to wholesalers and 8 to SDP. Are there special meaning to these?-MDS book chapter 22 has a great representation of distribution system (https://msh.org/wp-content/uploads/2013/04/mds3-ch22-distribution-mar2012.pdf). Check figure 22.1 on this book and see how you might modify this picture to make it clear as in the current state is not.4. in the methodology section 2.2: what was the main reason of getting data for 8 months before covid and 12 months during covid? This might have biased the data by skewing it towards one side. It is evident that there were more stock outs during covid, but no explanation has been done on the influence of the differences in the period of data collection. Please address this.5. table 1. include that the word "average days out of stock per month". Did you perform any test as to whether the results were
--

	statistically significant or not? Were there 95% confidence intervals performed on the average days out of stock?
REVIEWER	Hashim, Hashim Talib College of Medicine
REVIEW RETURNED	21-Nov-2023
GENERAL COMMENTS	It is a good paper.

VERSION 1 – AUTHOR RESPONSE

Reviewer 1

Thank you for a great article that has compared drug availability before and during COVID-19 pandemic. There are few articles in this field and will enhance measures aimed at promoting drug availability. However, the authors need to address the following:

We appreciate your interest in the topic and are glad to hear that you found it informative. We definitely took your suggestions into consideration and worked to address any concerns and suggestions as follows:

1. Paragraph two of the introduction section: Please replace the word "counterfeit". This word is no longer recommended by WHO as it is property rights issue. And use the word "falsified" as advocated by WHO.

♣ We appreciate your input, and the word is replaced to make sure to update the language in the article to reflect the WHO's recommendation to use the term "falsified" instead of "counterfeit". (Page 5, line 95)

2. diagram 1 is a general distribution system of Ethiopia. I would rather request the authors highlight the hubs under which the study took place than a picture of the whole country.

♣ Thank you for your feedback.. Hence, those branches/hubs were identified in black dot in the main manuscript (page 8)

3. Diagram 2. the diagram is slightly confusing in the following ways:

↳ It is assuming that Donors do manufacture medicines. this is not entirely true as in most cases they just finance the purchase of medicines from reputable supplies. So, the diagram has combined flow of product and funding. It needs to be corrected.

↳ What are the meaning of the arrows in diagram 2? I see nine arrows from API to manufacturers and four from Manufacturers to wholesalers and 8 to SDP. Are there special meaning to these?

↳ MDS book chapter 22 has a great representation of distribution system (<https://msh.org/wp-content/uploads/2013/04/mds3-ch22-distribution-mar2012.pdf>). Check figure 22.1 on this book and see how you might modify this picture to make it clear as in the current state is not.

♣ Dear, Thank you again for your valuable input. You are correct that donors do not manufacture medicines, but rather finance their purchase from reputable suppliers. We apologize for any confusion this may have caused. The diagram was intended to provide a general overview of the distribution system, including the flow of both products and funding. Regarding the arrows in Diagram 2, they represent the flow of products from one stage of the distribution system to another. The arrows from API to manufacturers indicate the flow of active pharmaceutical ingredients (APIs) from suppliers to manufacturers, while the arrows from manufacturers to wholesalers represent the flow of finished products from manufacturers to wholesalers. The arrows to SDPs represent the flow of products from wholesalers to service delivery points (SDPs), such as hospitals and clinics. Considering the valid comments raised by the reviewer, we have used the supply chain distribution system developed by MSH, with some modification considering the current Ethiopian pharmaceutical supply chain map with reference (page 10)

4. in the methodology section 2.2: what was the main reason of getting data for 8 months before covid and 12 months during covid? This might have biased the data by skewing it towards one side. It is

evident that there were more stock outs during covid, but no explanation has been done on the influence of the differences in the period of data collection. Please address this.

♣ Thank you for your comment and for bringing up an important point.. We acknowledge that this approach may have introduced some bias in the data, as the longer period of data collection during COVID-19 may have skewed the results. We have pointed out the reason and made revisions accordingly under methods section (section 2.2) line 134-137 using the following statement. "This period was purposefully selected. In Ethiopia, the month of May/June marks the culmination of fiscal year activities by the government and the preparation of the New Year budget. Consequently, we have only eight months counting back from December 31st, 2019 (the outbreak of COVID-19) to obtain the most up-to-date information about the stock."

5. table 1. include that the word "average days out of stock per month". Did you perform any test as to whether the results were statistically significant or not? Were there 95% confidence intervals performed on the average days out of stock?

♣ Thank you for your comment and for your suggestion regarding the inclusion of the term "average days out of stock per month" in Table 1 and revised accordingly.

♣ Regarding your question about statistical significance, we did statistical tests to compare the stockout rates before and during COVID-19, using two-sample t-test to compare the mean stockout rates, and mean percentage of availability between the two periods, and we also calculated 95% confidence intervals for the mean differences in stockout rates, as well as the percent of availability. These results are presented in the results section of the article Section 3.2, page 15, line 200-209. In addition, we have indicated this result in the Abstract section too.

Reviewer: 2

1. It is a good paper.

♣ Dear Reviewer,

Thank you for taking the time to review our article. We appreciate your positive feedback and are glad to hear that you found it well-written and nice.

VERSION 2 – REVIEW

REVIEWER	Khuluza, Felix University of Malawi, College of Medicine-Pharmacy Department
REVIEW RETURNED	18-Dec-2023

GENERAL COMMENTS	The authors have responded all previous queries that I raised. Minor edits that need to be checked are: 1. In the abstract, add a statement in the results section regarding unit price hike. This is because the issue of price hike has been highlighted in the conclusion 2. For key words, add "Ethiopia" 3. In the abstract: . "Furthermore, the mean number of days out of stock was 11.7 (95% CI: 9.9-13.5) before the outbreak of COVID-19"... Please state whether the days out of stock is per month or for the entire period.
---

VERSION 2 – AUTHOR RESPONSE

2. Reviewer: 1 The authors have responded to all previous queries that I raised. Minor edits that need to be checked are:

a) In the abstract, add a statement in the results section regarding unit price hike. This is because the issue of price hike has been highlighted in the conclusion.

♣ Dear Reviewer, thank you so much. As price changes was also one of the variables studied, we have included under result section of abstract accordingly.

b) For key words, add "Ethiopia."

♣ The word "Ethiopia" is included under keywords.

c) In the abstract. "Furthermore, the mean number of days out of stock was 11.7 (95% CI: 9.9-13.5) before the outbreak of COVID-19"... Please state whether the days out of stock is per month or for the entire period.

♣ Yes, it was the days out of stock per month. It is revised and all the revisions are highlighted in light blue color.